# Variational Temporal Abstraction

**Taesup Kim**[1,3,†], **Sungjin Ahn**[2*], **Yoshua Bengio**[1*]

[1]Mila, Université de Montréal, [2]Rutgers University, [3]Kakao Brain

## Abstract

We introduce a variational approach to learning and inference of temporally hierarchical structure and representation for sequential data. We propose the *Variational Temporal Abstraction* (VTA), a hierarchical recurrent state space model that can infer the latent temporal structure and thus perform the stochastic state transition hierarchically. We also propose to apply this model to implement the jumpy imagination ability in imagination-augmented agent-learning in order to improve the efficiency of the imagination. In experiments, we demonstrate that our proposed method can model 2D and 3D visual sequence datasets with interpretable temporal structure discovery and that its application to jumpy imagination enables more efficient agent-learning in a 3D navigation task.

## 1 Introduction

Discovering temporally hierarchical structure and representation in sequential data is the key to many problems in machine learning. In particular, for an intelligent agent exploring an environment, it is critical to learn such spatio-temporal structure hierarchically because it can, for instance, enable efficient option-learning and jumpy future imagination, abilities critical to resolving the sample efficiency problem (Hamrick, 2019). Without such temporal abstraction, imagination would easily become inefficient; imagine a person planning one-hour driving from her office to home with future imagination at the scale of every second. It is also biologically evidenced that future imagination is the very fundamental function of the human brain (Mullally & Maguire, 2014; Buckner, 2010) which is believed to be implemented via hierarchical coding of the grid cells (Wei et al., 2015).

There have been approaches to learn such hierarchical structure in sequences such as the HM-RNN (Chung et al., 2016). However, as a deterministic model, it has the main limitation that it cannot capture the stochastic nature prevailing in the data. In particular, this is a critical limitation to imagination-augmented agents because exploring various possible futures according to the uncertainty is what makes the imagination meaningful in many cases. There have been also many probabilistic sequence models that can deal with such stochastic nature in the sequential data (Chung et al., 2015; Krishnan et al., 2017; Fraccaro et al., 2017). However, unlike HMRNN, these models cannot automatically discover the temporal structure in the data.

In this paper, we propose the Hierarchical Recurrent State Space Model (HRSSM) that combines the advantages of both worlds: it can discover the latent temporal structure (e.g., subsequences) while also modeling its stochastic state transitions hierarchically. For its learning and inference, we introduce a variational approximate inference approach to deal with the intractability of the true posterior. We also propose to apply the HRSSM to implement efficient *jumpy imagination* for imagination-augmented agents. We note that the proposed HRSSM is a *generic* generative sequence model that is not tied to the specific application to the imagination-augmented agent but can be applied to any sequential data. In experiments, on 2D bouncing balls and 3D maze exploration, we show that the proposed model

Correspondence to `taesup.kim@umontreal.ca` and `sungjin.ahn@rutgers.edu`

can model sequential data with interpretable temporal abstraction discovery. Then, we show that the model can be applied to improve the efficiency of imagination-augmented agent-learning.

The main contributions of the paper are:

1. We propose the Hierarchical Recurrent State Space Model (HRSSM) that is the first stochastic sequence model that discovers the temporal abstraction structure.
2. We propose the application of HRSSM to imagination-augmented agent so that it can perform efficient jumpy future imagination.
3. In experiments, we showcase the temporal structure discovery and the benefit of HRSSM for agent learning.

## 2 Proposed Model

### 2.1 Hierarchical Recurrent State Space Models

In our model, we assume that a sequence $X = x_{1:T} = (x_1, \ldots, x_T)$ has a latent structure of temporal abstraction that can partition the sequence into $N$ non-overlapping subsequences $X = (X_1, \ldots, X_N)$. A subsequence $X_i = x^i_{1:l_i}$ has length $l_i$ such that $T = \sum_{i=1}^{T} l_i$ and $L = \{l_i\}$. Unlike previous works (Serban et al., 2017), we treat the number of subsequences $N$ and the lengths of subsequences $L$ as discrete latent variables rather than given parameters. This makes our model discover the underlying temporal structure adaptively and stochastically.

We also assume that a subsequence $X_i$ is generated from a *temporal abstraction $z_i$* and an observation $x_t$ has *observation abstraction $s_t$*. The temporal abstraction and observation abstraction have a hierarchical structure in such a way that all observations in $X_i$ are governed by the temporal abstraction $z_i$ in addition to the local observation abstraction $s_t$. As a temporal model, the two abstractions take temporal transitions. The transition of temporal abstraction occurs only at the subsequence scale while the observation transition is performed at every time step. This generative process can then be written as follows:

$$p(X, S, L, Z, N) = p(N) \prod_{i=1}^{N} p(X_i, S_i | z_i, l_i) p(l_i | z_i) p(z_i | z_{<i}) \tag{1}$$

where $S = \{s^i_j\}$ and $Z = \{z_i\}$ and the subsequence joint distribution $p(X_i, S_i | z_i, l_i)$ is:

$$p(X_i, S_i | z_i, l_i) = \prod_{j=1}^{l_i} p(x^i_j | s^i_j) p(s^i_j | s^i_{<j}, z_i). \tag{2}$$

We note that it is also possible to use the traditional Markovian state space model in Eqn. (1) and Eqn. (2) which has some desirable properties such as modularity and interpretability as well as having a closed-form solution for a limited class of models like the linear Gaussian model. However, it is known that this Markovian model has difficulties in practice in capturing complex long-term dependencies (Auger-Méthé et al., 2016). Thus, in our model, we take the recurrent state space model (RSSM) approach (Zheng et al., 2017; Buesing et al., 2018; Hafner et al., 2018b) which resolves this problem by adding a deterministic RNN path that can effectively encode the complex nonlinear long-term dependencies in the past, i.e., $z_{<i}$ and $s^i_{<j}$ in our model. Specifically, the transition is performed by the following updates: $c_i = f_{z\text{-rnn}}(z_{i-1}, c_{i-1})$, $z_i \sim p(z_i | c_i)$ for $z_i$, and $h^i_j = f_{s\text{-rnn}}(s^i_{j-1} || z_i, h^i_{j-1})$, $s^i_j \sim p(s^i_j | h^i_j)$ for $s^i_j$.

### 2.2 Binary Subsequence Indicator

Although the above modeling intuitively explains the actual generation process, the discrete latent random variables $N$ and $\{l_i\}$—whose realization is an integer—raise difficulties in learning and inference. To alleviate this problem, we reformulate the model by replacing the integer latent variables by a sequence of binary random variables $M = m_{1:T}$, called the *boundary indicator*. As the name implies, the role of this binary variable is to indicate whether a new subsequence should start at the next time step or not. In other words, it specifies the end of a subsequence. This is a similar operation to the FLUSH operation in the HMRNN model (Chung et al., 2016). With the binary indicators, the generative process can be rewritten as follows:

$$p(X, Z, S, M) = \prod_{t=1}^{T} p(x_t | s_t) p(m_t | s_t) p(s_t | s_{<t}, z_t, m_{t-1}) p(z_t | z_{<t}, m_{t-1})$$

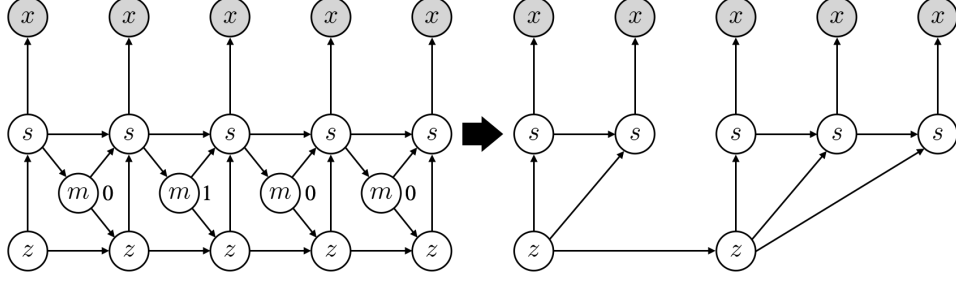

Figure 1: Sequence generative procedure (recurrent deterministic paths are excluded). **Left**: The model with the boundary indicators $M = \{0, 1, 0, 0\}$. **Right**: The corresponding generative procedure with a temporal structure derived from the boundary indicators $M$

In this representation of the generative process, we can remove the subsequence hierarchy and make both transitions perform at every time step. Although this seemingly looks different to our original generation process, the control of the binary indicator—selecting either COPY or UPDATE—can make this equivalent to the original generation process, which we explain later in more detail. In Figure 1, we provide an illustration on how the binary indicators induce an equivalent structure represented by the discrete random variables $N$ and $L$.

This reformulation has the following advantages. First, we do not need to treat the two different types of discrete random variables $N$ and $L$ separately but instead can unify them by using only one type of random variables $M$. Second, we do not need to deal with the variable range of $N$ and $L$ because each time step has finite states $\{0, 1\}$ while $N$ and $L$ depend on $T$ that can be changed across sequences. Lastly, the decision can be made adaptively while observing the progress of the subsequence, instead of making a decision governing the whole subsequence.

### 2.3 Prior on Temporal Structure

We model the binary indicator $p(m_t|s_t)$ by a Bernoulli distribution parameterized by $\sigma(f_{m\text{-mlp}}(s_t))$ with a multi-layer perceptron (MLP) $f_{m\text{-mlp}}$ and a sigmoid function $\sigma$. In addition, it is convenient to explicitly express our prior knowledge or constraint on the temporal structure using the boundary distribution. For instance, it is convenient to specify the maximum number of subsequences $N_{\max}$ or the longest subsequence lengths $l_{\max}$ when we do not want too many or too long subsequences. To implement, at each time step $t$, we can compute the number of subsequences discovered so far by using a counter $n(m_{<t})$ as well as the length of current subsequence with another counter $l(m_{<t})$. Based on this, we can design the boundary distribution with our prior knowledge as follows:

$$p\left(m_t = 1|s_t\right) = \begin{cases} 0 & \text{if } n\left(m_{<t}\right) \geq N_{\max}, \\ 1 & \text{elseif } l\left(m_{<t}\right) \geq l_{\max}, \\ \sigma\left(f_{m\text{-mlp}}\left(s_t\right)\right) & \text{otherwise.} \end{cases}$$

### 2.4 Hierarchical Transitions

The transition of temporal abstraction should occur only a subsequence is completed. This timing is indicated by the boundary indicator. Specifically, the transition of temporal abstraction is implemented as follows:

$$p\left(z_t|z_{<t}, m_{t-1}\right) = \begin{cases} \delta(z_t = z_{t-1}) & \text{if } m_{t-1} = 0 \text{ (COPY)}, \\ \tilde{p}(z_t|c_t) & \text{otherwise (UPDATE)} \end{cases}$$

where $c_t$ is the following RNN encoding of all previous temporal abstractions $z_{<t}$ (and $m_{<t}$):

$$c_t = \begin{cases} c_{t-1} & \text{if } m_{t-1} = 0 \text{ (COPY)}, \\ f_{z\text{-rnn}}\left(z_{t-1}, c_{t-1}\right) & \text{otherwise (UPDATE)}. \end{cases}$$

Specifically, having $m_{t-1} = 0$ indicates that the time step $t$ is still in the same subsequence as the previous time step $t - 1$ and thus the temporal abstraction should not be updated but copied. Otherwise, it indicates that the time step $t - 1$ was the end of the previous subsequence and thus the

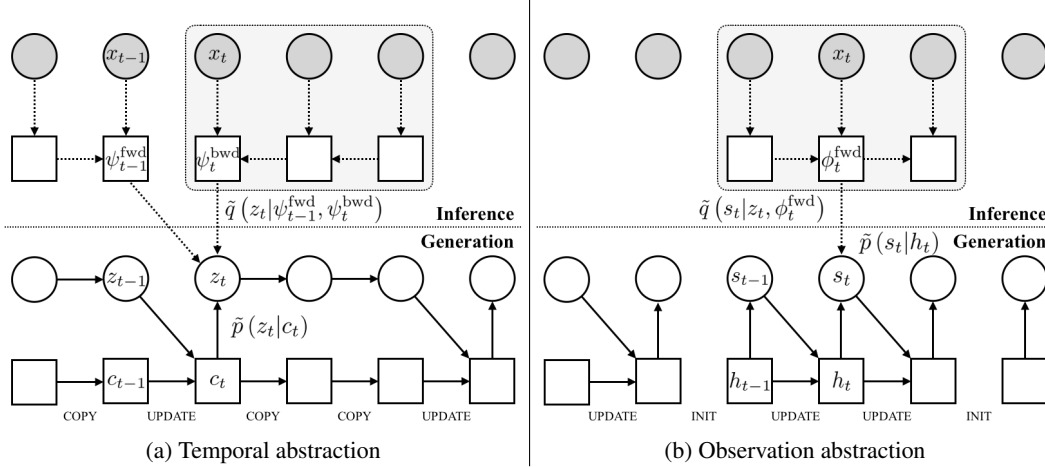

(a) Temporal abstraction           (b) Observation abstraction

Figure 2: State transitions: inference and generation with a given hierarchical temporal structure based on the boundary indicators $M$.

temporal abstraction should be updated. This transition is implemented as a Gaussian distribution $\mathcal{N}(z_t|\mu_z(c_t), \sigma_z(c_t))$ where both $\mu_z$ and $\sigma_z$ are implemented with MLPs.

At test time, we can use this transition of temporal abstraction without the COPY mode, i.e., every transition is UPDATE. This implements the *jumpy* future imagination which do not require to rollout at every raw time step and thus is computationally efficient.

The observation transition is similar to the transition of temporal abstraction except that we want to implement the fact that given the temporal abstraction $z_i$, a subsequence is independent of other subsequences. The observation transition is implemented as follows:

$$p(s_t|s_{<t}, z_t, m_{t-1}) = \tilde{p}(s_t|h_t) \quad \text{where} \quad h_t = \begin{cases} f_{\text{s-rnn}}(s_{t-1}\|z_t, h_{t-1}) & \text{if } m_{t-1} = 0 \text{ (UPDATE)}, \\ f_{\text{s-mlp}}(z_t) & \text{otherwise (INIT)}. \end{cases}$$

Here, $h_t$ is computed by using an RNN $f_{\text{s-rnn}}$ to update (UPDATE), and a MLP $f_{\text{s-mlp}}$ to initialize (INIT). The concatenation is denoted by $\|$. Note that if the subsequence is finished, i.e., $m_{t-1} = 1$, we sample a new observational abstraction $s_t$ without conditioning on $h_t$. That is, the underlying RNN is initialized.

# 3 Learning and Inference

As the true posterior is intractable, we apply variational approximation which gives the following evidence lower bound (ELBO) objective:

$$\log p(X) \geq \sum_M \int_{Z,S} q_\phi(Z, S, M|X) \log \frac{p_\theta(X, Z, S, M)}{q_\phi(Z, S, M|X)} dZ dS$$

This is optimized w.r.t. $\theta$ and $\phi$ using the reparameterization trick (Kingma & Welling, 2014). In particular, we use the Gumbel-softmax (Jang et al., 2017; Maddison et al., 2017) with straight-through estimators (Bengio et al., 2013) for the discrete variables $M$. For the approximate posterior, we use the following factorization:

$$q_\phi(Z, S, M|X) = q(M|X)q(Z|M, X)q(S|Z, M, X).$$

That is, by *sequence decomposition* $q(M|X)$, we first infer the boundary indicators independent of $Z$ and $S$. Then, given the discovered boundary structure, we infer the two abstractions via the *state inference* $q(Z|M, X)$ and $q(S|Z, M, X)$.

**Sequence Decomposition.** Inferring the subsequence structure is important because the other state inference can be decomposed into independent subsequences. This sequence decomposition is implemented by the following decomposition:

$$q(M|X) = \prod_{t=1}^T q(m_t|X) = \prod_{t=1}^T \text{Bern}(m_t|\sigma(\varphi(X))),$$

where $\varphi$ is a convolutional neural network (CNN) applying convolutions over the temporal axis to extract dependencies between neighboring observations. This enables to sample all indicators $M$ independently and simultaneously. Empirically, we found this CNN-based architecture working better than an RNN-based architecture.

**State Inference.** State inference is also performed hierarchically. The temporal abstraction predictor $q(Z|M, X) = \prod_{t=1}^{T} q(z_t|M, X)$ does inference by encoding subsequences determined by $M$ and $X$. To use the same temporal abstraction across the time steps of a subsequence, the distribution $q(z_t|M, X)$ is also conditioned on the boundary indicator $m_{t-1}$:

$$q(z_t|M, X) = \begin{cases} \delta(z_t = z_{t-1}) & \text{if } m_{t-1} = 0 \text{ (COPY)}, \\ \tilde{q}(z_t|\psi_{t-1}^{\text{fwd}}, \psi_t^{\text{bwd}}) & \text{otherwise (UPDATE)}. \end{cases}$$

We use the distribution $\tilde{q}(z_t|\psi_{t-1}^{\text{fwd}}, \psi_t^{\text{bwd}})$ to update the state $z_t$. It is conditioned on all previous observations $x_{<t}$ and this is represented by a feature $\psi_{t-1}^{\text{fwd}}$ extracted from a forward RNN $\psi^{\text{fwd}}(X)$. The other is a feature $\psi_t^{\text{bwd}}$ representing the current step's subsequence that is extracted from a backward (masked) RNN $\psi^{\text{bwd}}(X, M)$. In particular, this RNN depends on $M$, which is used as a masking variable, to ensure independence between subsequences.

The observation abstraction predictor $q(S|Z, M, X) = \prod_{t=1}^{T} q(s_t|z_t, M, X)$ is factorized and each observational abstraction $s_t$ is sampled from the distribution $q(s_t|z_t, M, X) = \tilde{q}(s_t|z_t, \phi_t^{\text{fwd}})$. The feature $\phi_t^{\text{fwd}}$ is extracted from a forward (masked) RNN $\phi^{\text{fwd}}(X, M)$ that encodes the observation sequence $X$ and resets hidden states when a new subsequence starts.

# 4 Related Works

The most similar work with our model is the HMRNN (Chung et al., 2016). While it is similar in the sense that both models discover the hierarchical temporal structure, HMRNN is a deterministic model and thus has a severe limitation to use for an imagination module. In the switching state-space model (Ghahramani & Hinton, 2000), the upper layer is a Hidden Markov Model (HMM) and the behavior of the lower layer is modulated by the discrete state of the upper layer, and thus gives hierarchical temporal structure. Linderman et al. (2016) proposed a new class of switching state-space models that discovers the dynamical units and also explains the switching behavior depending on observations or continuous latent states. The authors used inference based on message-passing. The hidden semi-Markov models (Yu, 2010; Dai et al., 2016) perform similar segmentation with discrete states. However, unlike our model, there is no states for temporal abstraction. Kipf et al. (2018) proposed soft-segmentation of sequence for compositional imitation learning.

The variational recurrent neural networks (VRNN) (Chung et al., 2015) is a latent variable RNN but uses auto-regressive state transition taking inputs from the observation. Thus, this can be computationally expensive to use as an imagination module. Also, the error can accumulate more severely in the high dimensional rollout. To resolve this problem, Krishnan et al. (2017) and Buesing et al. (2018) proposes to combine the traditional Markovian State Space Models with deep neural networks. Zheng et al. (2017) and Hafner et al. (2018a) proposed to use an RNN path to encode the past making non-Markovian state-space models which can alleviate the limitation of the traditional SSMs. Serban et al. (2017) proposed a hierarchical version of VRNN called Variational Hierarchical Recurrent Encoder-Decoder (VHRED) which results in a similar model as ours. However, it is a significant difference that our model learns the segment while VHRED uses a given structure. A closely related work is TDVAE (Gregor et al., 2019). TDVAE is trained on pairs of temporally separated time points. Jayaraman et al. (2019) and Neitz et al. (2018) proposed models that predict the future frames that, unlike our approach, have the lowest uncertainty. The resulting models predict a small number of easily predictable "bottleneck" frames through which any possible prediction must pass. Pertsch et al. (2019) proposed to predict the keyframes with their temporal offsets using stochastic prediction and deterministically interpolate the intermediate frames.

# 5 Experiments

We demonstrate our model on visual sequence datasets to show (1) how sequence data is decomposed into perceptually plausible subsequences without any supervision, (2) how jumpy future prediction is

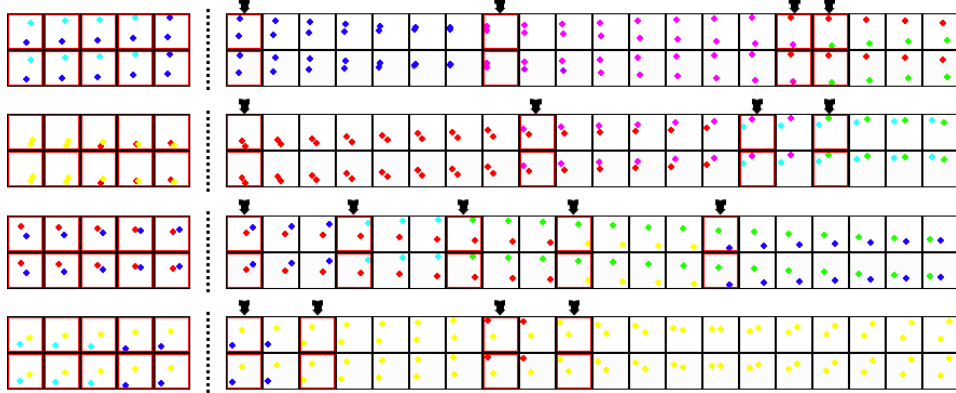

Figure 3: **Left**: Previously observed (context) data $X_{\text{ctx}}$. **Right**: Each first row is the input observation sequence $X$ and the second row is the corresponding reconstruction. The sequence decomposer $q(M|X)$ predicts the starting frames of subsequences and it is indicated by arrows (red squared frames). Subsequences are newly defined when a ball hits the wall by changing the color.

done with temporal abstraction and (3) how this jumpy future prediction can improve the planning as an imagination module in a navigation problem. Moreover, we test conditional generation $p(X|X_{\text{ctx}})$ where $X_{\text{ctx}} = x_{-(T_{\text{ctx}}-1):0}$ is the context observation of length $T_{\text{ctx}}$. With the context, we preset the state transition of the temporal abstraction by deterministically initializing $c_0 = f_{\text{ctx}}(X_{\text{ctx}})$ with $f_{\text{ctx}}$ implemented by a forward RNN. The code of the implementation of our model is available at https://github.com/taesupkim/vta.

## 5.1 Bouncing Balls

We generated a synthetic 2D visual sequence dataset called *bouncing balls*. The dataset is composed of two colored balls that are designed to bounce in hitting the walls of a square box. Each ball is independently characterized with certain rules: (1) The color of each ball is randomly changed when it hits a wall and (2) the velocity (2D vector) is also slightly changed at every time steps with a small amount of noise. We trained a model to learn 1D state representations and all observation data $x_t \in \mathbb{R}^{32 \times 32 \times 3}$ are encoded and decoded by convolutional neural networks. During training, the length of observation sequence data $X$ is set to $T = 20$ and the context length is $T_{\text{ctx}} = 5$. Hyper-parameters related to sequence decomposition are set as $N_{\max} = 5$ and $l_{\max} = 10$.

Our results in Figure 3 show that the sequence decomposer $q(M|X)$ predicts reasonable subsequences by setting a new subsequence when the color of balls is changed or the ball is bounced. At the beginning of training, the sequence decomposer is unstable with having large entropy and tends to define subsequences with a small number of frames. It then began to learn to increase the length of subsequences and this is controlled by annealing the temperature $\tau$ of Gumbel-softmax towards small values from 1.0 to 0.1. However, without our proposed prior on temporal structure, the sequence decomposer fails to properly decompose sequences and our proposed model consequently converges into RSSM.

## 5.2 Navigation in 3D Maze

Another sequence dataset is generated from the 3D maze environment by an agent that navigates the maze. Each observation data $x_t \in \mathbb{R}^{32 \times 32 \times 3}$ is defined as a partially observed view observed by the agent. The maze consists of hallways with colored walls and is defined on a $26 \times 18$ grid map as shown in Figure 5. The agent is set to navigate around this environment and the viewpoint of the agent is constantly jittered with some noise. We set some constraints on the agent's action (*forward, left-turn, right-turn*) that the agent is not allowed to turn around when it is located on the hallway. However, it can turn around when it arrives nearby intersections between hallways. Due to these constraints, the agent without a policy can randomly navigate the maze environment and collect meaningful data. To train an environment model, we collected 1M steps (frames) from the randomly navigating agent and used it to train both RSSM and our proposed HRSSM. For HRSSM, we used the same training setting as bouncing balls but different $N_{\max} = 5$ and $l_{\max} = 8$ for the sequence decomposition. The corresponding learning curves are shown in Figure 4 that both reached a similar ELBO. This suggests that our model does not lose the reconstruction performance

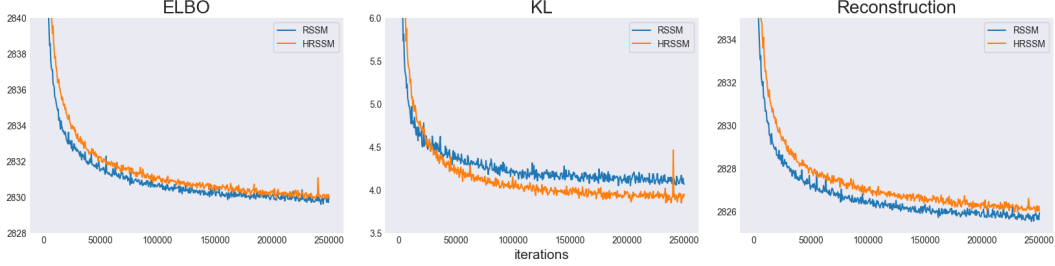

Figure 4: The learning curve of RSSM and HRSSM: ELBO, KL-divergence and reconstruction loss.

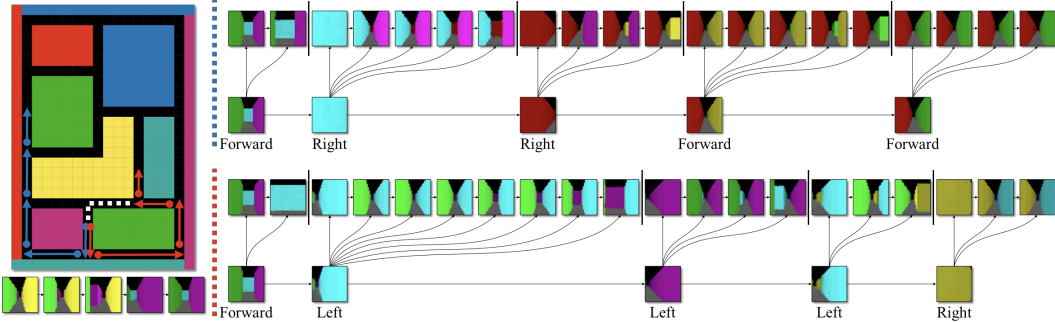

Figure 5: **Left**: Bird's-eye view (map) of the 3D maze with generated navigation paths. White dotted lines indicate the given context path $X_{ctx}$ and the corresponding frames are depicted below the map. Solid lines are the generated paths (blue: top, red: bottom) conditioned on the same context. Circles are the starting points of subsequences where the temporal abstract transitions exactly take place. **Right**: The generated sequence data is shown with its temporal structure. Both generations are conditioned on the same context but different input actions as indicated. Frame samples on each bottom row are generated with the temporal abstract transition $\tilde{p}\left(z_{t'}|c_{t'}\right)$ with $c_{t'} = f_{z\text{-rnn}}\left(z_{t'-1}, c_{t'-1}\right)$ and this shows how the jumpy future prediction is done. Other samples on top rows, which are not necessarily required for future prediction with our proposed HRSSM, are generated from the observation abstraction transition $\tilde{p}\left(s_t|h_t\right)$ with $h_t = f_{s\text{-rnn}}\left(s_{t-1}\|z_t, h_{t-1}\right)$. The boundaries between subsequences are determined by $p\left(m_t|s_t\right)$.

while discovering the hierarchical structure. We trained state transitions to be action-conditioned and therefore this allows to perform action-controlled imagination. For HRSSM, only the temporal abstraction state transition is action-conditioned as we aim to execute the imagination only with the jumpy future prediction. The overall sequence generation procedure is described in Figure 5. The temporal structure of the generated sequence shows how the jumpy future prediction works and where the transitions of temporal abstraction occur. We see that our model learns to set each hallway as a subsequence and consequently to perform jumpy transitions between hallways without repeating or skipping a hallway. In Figure 6, a set of jumpy predicted sequences from the same context $X_{ctx}$ and different input actions are shown and this can be interpreted as imaginations the agent can use for planning.

**Goal-Oriented Navigation**   We further use the trained model as an imagination module by augmenting it to an agent to perform the goal-oriented navigation. In this experiment, the task is to navigate to a randomly selected goal position within the given life steps. The goal position in the grid map is not provided to the agent, but a $3 \times 3$-cropped image around the goal position is given. To reach the goal fast, the agent is augmented with the imagination model and allowed to execute a rollout over a number of imagination trajectories (i.e., a sequence of temporal abstractions) by varying the input actions. Afterward, it decides the best trajectory that helps to reach the goal faster. To find the best trajectory, we use a simple strategy: a cosine-similarity based matching between all imagined state representations in imaginations and the feature of the goal image. The feature extractor for the goal image is jointly trained with the model. [2] This way, at every time step we let

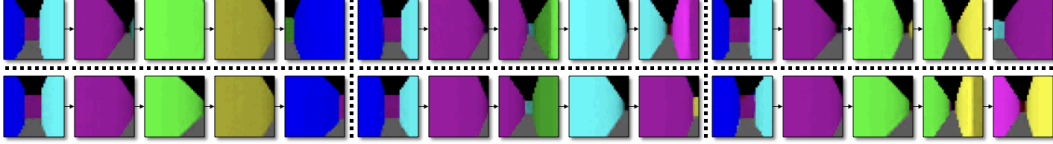

Figure 6: Jumpy future prediction conditioned on the same context $X_{\text{ctx}}$ and different input actions

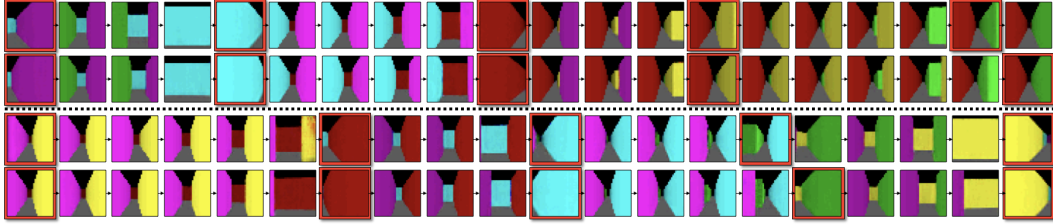

Figure 7: Fully generate seqences conditioned on the same context $X_{\text{ctx}}$ and same input actions: generated paths are equal but the viewpoint and the lengths of subsequences are varied (red squared frames are jumpy future predictions).

the agent choose the first action resulting in the best trajectory. This approach can be considered as a simple variant of the Monte Carlo Tree Search (MCTS) and the detailed overall procedure can be found in Appendix. Each episode is defined by randomly initializing the agent position and the goal position. The agent is allowed maximum 100 steps to reach the goal and the final reward is defined as the number of remaining steps when the agent reaches the goal or consumes all life-steps. The performance highly depends on the accuracy and the computationally efficiency of the model and we therefore compare between RSSM and HRSSM with varying the length of imagined trajectories. We measure the performance by randomly generated 5000 episodes and show how each setting performs across the episodes by plotting the reward distribution in Figure 8. It is shown that the HRSSM significantly improves the performance compared to the RSSM by having the same computational budget.

HRSSM showed consistent performance over different lengths of imagined trajectories and most episodes were solved within 50 steps. We believe that this is because HRSSM is able to abstract multiple time steps within a single state transition and this enables to reduce the computational cost for imaginations. The results also show that finding the best trajectory becomes difficult as the imagination length gets larger, i.e., the number of possible imagination trajectories increases. This suggests that imaginations with temporal abstraction can benefit both the accuracy and the computationally efficiency in effective ways.

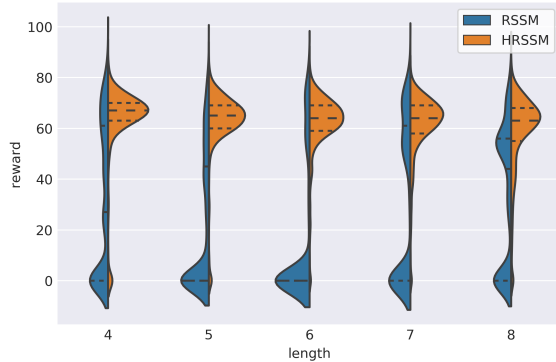

Figure 8: Goal-oriented navigation with different lengths of imagined trajectories.

## 6 Conclusion

In this paper, we introduce the *Variational Temporal Abstraction* (VTA), a generic generative temporal model that can discover the hierarchical temporal structure and its stochastic hierarchical state transitions. We also propose to use this temporal abstraction for temporally-extended future imagination in imagination-augmented agent-learning. Experiment results shows that in general sequential data modeling, the proposed model discovers plausible latent temporal structures and perform hierarchical stochastic state transitions. Also, in connection to the model-based imagination-augmented agent for a 3D navigation task, we demonstrate the potential of the proposed model in improving the efficiency of agent-learning.

**Acknowledgments**

We would like to acknowledge Kakao Brain cloud team for providing computing resources used in this work. TK would also like to thank colleagues at Mila, Kakao Brain, and Rutgers Machine Learning Group. SA is grateful to Kakao Brain, the Center for Super Intelligence (CSI), and Element AI for their support. Mila (TK and YB) would also like to thank NSERC, CIFAR, Google, Samsung, Nuance, IBM, Canada Research Chairs, Canada Graduate Scholarship Program, and Compute Canada.

## Footnotes

*Equal advising, †work also done while visiting Rutgers University.

[2]During training, the $3 \times 3$ window (image) around the agent position is always given as additional observation data and we trained feature extractor by maximizing the cosine-similarity between the extracted feature and the corresponding time step state representation.

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
