[Supplementary Material]

## Appendix A  Action-Conditioned Temporal Abstraction State Transition

We implement action-conditioned state transition

$$p\left(z_t | a_t, z_{<t}, m_{<t}\right) = \begin{cases} \delta(z_t = z_{t-1}) & \text{if } m_{t-1} = 0 \text{ (COPY)}, \\ \tilde{p}(z_t | c_t) & \text{otherwise (UPDATE)} \end{cases}$$

where the action input $a_t$ is only affecting the UPDATE operation and we feed it into the deterministic path as the following:

$$c_t = \begin{cases} c_{t-1} & \text{if } m_{t-1} = 0 \text{ (COPY)}, \\ f_z\left(z_{t-1} \| a_t, c_{t-1}\right) & \text{otherwise (UPDATE)}. \end{cases}$$

## Appendix B  Goal-Oriented Navigation

---

**Algorithm 1** Goal-oriented navigation with imagination (single episode)

---

**Input:** environment `env`, environment model $\mathcal{E}$, maximum length of imagined trajectories $l_{\text{img}}$
**Output:** reward $r$

 1: Initialize reward $r \leftarrow 100$
 2: Initialize environment $x \leftarrow$ `env.reset()`
 3: Initialize context $X_{\text{ctx}} \leftarrow [x]$
 4: Sample goal position and extract goal map feature $g$
 5: **while** $r > 0$ **do**
 6:    Sample action $a$ from imagination-based planner given $\mathcal{E}, X_{\text{ctx}}, g, l_{\text{img}}$ (Algorithm 2)
 7:    Do action $x \leftarrow$ `env.step`$(a)$
 8:    Update context $X_{\text{ctx}} \leftarrow X_{\text{ctx}} + [x]$
 9:    Update reward $r \leftarrow r - 1$
10:    **if** current position is at the goal position **then**
11:        **break**
12: **return** reward $r$

---

**Algorithm 2** Imagination-based planner

---

**Input:** environment model $\mathcal{E}$, previously observed sequence (context) $X_{\text{ctx}}$, maximum length of imagined trajectories $l_{\text{img}}$, goal map feature $g$
**Output:** action $a_{\text{max}}$

 1: Initialize model $\mathcal{E}$ with $c_0 = f_{\text{ctx}}\left(X_{\text{ctx}}\right)$
 2: Initialize $d_{\text{max}} \leftarrow -\infty$
 3: Initialize $a_{\text{max}} \leftarrow$ None
 4: Set a list $\mathcal{A}$ of all possible action sequences based on $l_{\text{img}}$
 5: **for** each action seuqence $A$ in $\mathcal{A}$ **do**
 6:    Get a sequence of states $\mathcal{S} \in \mathbb{R}^{l_{\text{img}} \times d}$ by doing imagination with model $\mathcal{E}$ conditioned on $A$
 7:    Compute cosine-similarity $D \in \mathbb{R}^{l_{\text{img}}}$ between all states $\mathcal{S}$ and goal map feature $g$
 8:    **if** $\max(D) > d_{\text{max}}$ **then**
 9:        Update $d_{\text{max}} \leftarrow \max(D)$
10:        Update $a_{\text{max}} \leftarrow A[0]$
11: **return** action $a_{\text{max}}$

---

## Appendix C  Implementation Details

For bouncing balls, we define the reconstruction loss (data likelihood) by using binary cross-entropy. The images from 3D maze are pre-processed by reducing the bit depth to 5 bits (Kingma & Dhariwal, 2018) and therefore the reconstruction loss is computed by using Gaussian distribution. We used the AMSGrad (Reddi et al., 2018), a variant of Adam, optimizer with learning rate $5e - 4$ and all mini-batchs are with 64 sequences with length $T = 20$. Both CNN-based encoder and decoder are composed of 4 convolution layers with ELU activations (Clevert et al., 2016). A GRU (Cho et al., 2014) is used for all RNNs with 128 hidden units. The state representations of temporal abstraction and observation abstraction are sampled from 8-dimensional diagional Gaussian distributions.

## Appendix D  Evidence Lower Bound (ELBO)

We derive the ELBO without considering recurrent deterministic paths.

**Log-likelihood** $\log p\left(X\right)$

$$
\begin{aligned}
\log p\left(X\right) &= \log \sum_M \int_{Z,S} p\left(X,Z,S,M\right) \\
&\geq \sum_M \int_{Z,S} q\left(Z,S,M|X\right) \log \frac{p\left(X,Z,S,M\right)}{q\left(Z,S,M|X\right)} \\
&= \sum_M \int_{Z,S} q\left(Z,S,M|X\right) \log \frac{p\left(X|Z,S\right)p\left(Z,S,M\right)}{q\left(Z,S,M|X\right)} \\
&= \sum_M \int_{Z,S} q\left(Z,S,M|X\right) \left[\log p\left(X|Z,S\right) + \log \frac{p\left(Z,S,M\right)}{q\left(Z,S,M|X\right)}\right] \\
&= \underbrace{\mathbb{E}_{q(Z,S,M|X)}\left[\log p\left(X|Z,S\right)\right]}_{\text{reconstruction}} - \underbrace{\text{KL}\left[q\left(Z,S,M|X\right)||p\left(Z,S,M\right)\right]}_{\text{KL divergence}}
\end{aligned}
$$

**Decomposing** $p\left(X|Z,S\right)$ **and** $p\left(Z,S,M\right)$

$$
p\left(X|Z,S\right) = \prod_t \underbrace{p\left(x_t|s_t\right)}_{\text{decoder}}
$$

$$
p\left(Z,S,M\right) = \prod_t \underbrace{p\left(z_t|z_{t-1},m_{t-1}\right)}_{\text{temporal abstract transition}} \underbrace{p\left(s_t|s_{t-1},z_t,m_{t-1}\right)}_{\text{observation abstract transition}} \underbrace{p\left(m_t|s_t\right)}_{\text{boundary prior}}
$$

**Decomposing** $q\left(Z,S,M|X\right)$

$$
\begin{aligned}
q\left(Z,S,M|X\right) &= q\left(M|X\right)q\left(Z,S|M,X\right) \\
&= q\left(M|X\right)q\left(Z|M,X\right)q\left(S|Z,M,X\right) \\
&= q\left(M|X\right)\prod_t q\left(z_t|M,X\right)q\left(s_t|z_t,M,X\right) \\
q(M|X) &= \prod_t q(m_t|X) = \prod_t \text{Bern}\left(m_t|\sigma(\varphi(X))\right)
\end{aligned}
$$

**Reconstruction Term in ELBO**

$$
\begin{aligned}
&\sum_M \int_{Z,S} q\left(Z,S,M|X\right) \log p\left(X|Z,S\right) \\
&= \underbrace{\sum_M q\left(M|X\right)}_{\text{sample } M} \int_{Z,S} q\left(Z,S|M,X\right) \log p\left(X|Z,S\right) \\
&\approx \int_{Z,S} q\left(Z,S|M,X\right) \sum_t \log p\left(x_t|s_t\right) \\
&= \int_{Z,S} \sum_t q\left(z_t,s_t|M,X\right) \log p\left(x_t|s_t\right) \\
&= \int_{Z,S} \sum_t \underbrace{q\left(z_t|M,X\right)q\left(s_t|z_t,M,X\right)}_{\text{sampling } z_t \text{ and } s_t} \log p\left(x_t|z_t,s_t\right) \\
&\approx \sum_t \log p\left(x_t|s_t\right)
\end{aligned}
$$

**KL Term in ELBO**

$$\log q\left(Z, S, M | X\right) - \log p\left(Z, S, M\right)$$

$$= \log q\left(M | X\right) + \log q\left(Z, S | M, X\right) - \log p\left(Z, S, M\right)$$

$$= \log q\left(M | X\right) + \log q\left(Z | M, X\right) + \log q\left(S | Z, M, X\right) - \log p\left(Z, S, M\right)$$

$$= \log q\left(M | X\right) + \sum_t \log \frac{q\left(z_t | M, X\right) q\left(s_t | z_t, M, X\right)}{p\left(z_t | z_{t-1}, m_{t-1}\right) p\left(s_t | s_{t-1}, z_t, m_{t-1}\right) p\left(m_t | s_t\right)}$$

$$= \log q\left(M | X\right) + \sum_t \log \frac{q\left(z_t | M, X\right)}{p\left(z_t | z_{t-1}, m_{t-1}\right)} + \log \frac{q\left(s_t | z_t, M, X\right)}{p\left(s_t | s_{t-1}, z_t, m_{t-1}\right)} + \log \frac{1}{p\left(m_t | s_t\right)}$$

$$\sum_M \int_{Z,S} q\left(Z, S, M | X\right) \left[\log q\left(Z, S, M | X\right) - \log p\left(Z, S, M\right)\right]$$

$$= \sum_M q\left(M | X\right) \int_{Z,S} q\left(Z, S | M, X\right) \left[\log q\left(Z, S, M | X\right) - \log p\left(Z, S, M\right)\right]$$

$$= \sum_M q\left(M | X\right) \int_{Z,S} q\left(Z, S | M, X\right) \left[\log q\left(M | X\right) + \sum_t \log \frac{q\left(z_t | M, X\right) q\left(s_t | z_t, M, X\right)}{p\left(z_t | z_{t-1}, m_{t-1}\right) p\left(s_t | s_{t-1}, z_t, m_{t-1}\right) p\left(m_t | s_t\right)}\right]$$

$$= \sum_M q\left(M | X\right) \left[\log q\left(M | X\right) + \sum_t \int_{Z,S} q\left(Z, S | M, X\right) \log \frac{q\left(z_t | M, X\right) q\left(s_t | z_t, M, X\right)}{p\left(z_t | z_{t-1}, m_{t-1}\right) p\left(s_t | s_{t-1}, z_t, m_{t-1}\right) p\left(m_t | s_t\right)}\right]$$

$$= \sum_M q\left(M | X\right) \left[\log q\left(M | X\right) + \sum_t \int_{z_t,s_t} q\left(z_t, s_t | M, X\right) \log \frac{q\left(z_t | M, X\right) q\left(s_t | z_t, M, X\right)}{p\left(z_t | z_{t-1}, m_{t-1}\right) p\left(s_t | s_{t-1}, z_t, m_{t-1}\right) p\left(m_t | s_t\right)}\right]$$

$$= \sum_M q\left(M | X\right) \left[\log q\left(M | X\right) + \sum_t \text{KL}\left(q'(z_t) || p'(z_t)\right) + \int_{z_t,s_t} q\left(z_t, s_t | M, X\right) \log \frac{q\left(s_t | z_t, M, X\right)}{p\left(s_t | s_{t-1}, z_t, m_{t-1}\right) p\left(m_t | s_t\right)}\right]$$

$$= \sum_M q\left(M | X\right) \left[\log q\left(M | X\right) + \sum_t \text{KL}\left(q'(z_t) || p'(z_t)\right) + \int_{z_t} \underbrace{q'\left(z_t\right)}_{\text{sample } z_t} \left[\text{KL}\left(q'(s_t) || p'(s_t)\right) - \log p\left(m_t | s_t\right)\right]\right]$$

$$\approx \sum_M q\left(M | X\right) \left[\log q\left(M | X\right) + \sum_t \text{KL}\left(q'(z_t) || p'(z_t)\right) + \text{KL}\left(q'(s_t) || p'(s_t)\right) - \log p\left(m_t | s_t\right)\right]$$

$$= \sum_M q\left(M | X\right) \left[\log \frac{\prod_t q\left(m_t | X\right)}{\prod_t p\left(m_t | s_t\right)} + \sum_t \text{KL}\left(q'(z_t) || p'(z_t)\right) + \text{KL}\left(q'(s_t) || p'(s_t)\right)\right]$$

$$= \sum_{t'} \text{KL}\left(q'\left(m_{t'}\right) || p'\left(m_{t'}\right)\right) + \sum_M \underbrace{q\left(M | X\right)}_{\text{sample } M} \left[\sum_t \text{KL}\left(q'\left(z_t\right) || p'\left(z_t\right)\right) + \text{KL}\left(q'\left(s_t\right) || p'\left(s_t\right)\right)\right]$$

$$\approx \sum_t \underbrace{\text{KL}\left(q'\left(m_t\right) || p'\left(m_t\right)\right)}_{\text{sequence decomposer}} + \underbrace{\text{KL}\left(q'\left(z_t\right) || p'\left(z_t\right)\right)}_{\text{temporal abstraction}} + \underbrace{\text{KL}\left(q'\left(s_t\right) || p'\left(s_t\right)\right)}_{\text{observation abstraction}}$$

where $p'(m_t) = p(m_t | s_t), p'(z_t) = p\left(z_t | z_{t-1}, m_{t-1}\right), p'(s_t) = p\left(s_t | s_{t-1}, z_t, m_{t-1}\right)$
$q'(m_t) = q(m_t | X), q'(z_t) = q\left(z_t, s_t | M, X\right), q'(s_t) = q\left(s_t | z_t, M, X\right)$

# Appendix E  Generated Samples

(a) Subsequences with different $z$ (and different $s$)

(b) Subsequences with same $z$ (and different $s$)

Figure 1: Generated subsequences with different content and temporal structure