[Reviews · NeurIPS 2019]

Reviewer 1



Weakness: My doubt mainly lies in the experiments section. 1. There is not enough quantitative evaluation of the model. As the authors claim, the proposed framework should be able to capture long-term temporal dependencies. This ability would result in higher generative likelihood. However, there is not enough quantitative evaluation and comparison to back up this statement. 2. The latent space, especially the temporal abstraction level is not investigated enough. Since the proposed framework should be able to learn high-hierarchical temporal structures, it would be interesting to traverse the temporal abstraction latent variable and visualize what happens. Does it encode different information with the observation abstraction or are they somehow entangled together? Such investigation would provide more insights into the hierarchical latent space learned. 3. Although it is not a big issue, the use of binary indicator with Gumbel-softmax relaxation has been utilized in a lot of previous works. But since it works, and it only serves as part of the contribution, I do not see it as a big issue.

Reviewer 2



1) Main main point of criticism is the experimental validation of the proposed model. 1.1) Sec 5.1: Bouncing balls (BBs) dataset 1.1.1) I think it is indeed good practice to test the algorithm on a simple dataset, but this version of BBs seems quite tailored to the algorithm, as the balls change color on collision. Does the segmentation still yield interpretable results without color change? 1.1.2) There is no quantitative comparison to an reasonable baseline model (eg matched for same network size or similar). This would be required to convince the reader that the inference and learning algorithms are able to identify the model. Also it would be good to see a sample from the baseline. 1.2) Sec 5.2: 3D Maze 1.2.1) My main quibble with this experiment is that the true segmentation is basically almost explicitly given based on the available actions, eg if TURN-LEFT is executed, then a new segment is allocated. This essentially points to the basic dilemma of hierarchical reinforcement learning: If I know good high-level options (here: always follow corridors to the next intersection) then learning the right, high-level state abstraction is easy; and vice-versa. Learning both at the same time is hard. I would be more convinced by these experiments if the authors ran an experiment eg with a model that's not conditioned on actions and see if segmentations still coincide with intersections. 1.2.2) How is the baseline RSSM defined here? How much do training curves vary across runs (let alone hyperparameters)? 2) Smaller comments: 2.1) Sec 2.3 l102-l103: This prior is quite weird as the last segment is different than the other ones. I don't really see the reason for this design choice, as the posterior inference does not make use of the maximum number of segments. 2.2) l135-l138: The assumption of independence of the $m_t$ under the posterior seems quite weak. Imagine in the BBs data set (no color change) it could be quite hard to determine where exactly the change point is (collision), but we can be very certain that there is only one. This situation could not be represented well with an independent posterior. 2.3) l40-l41: Clearly there have be earlier "stochastic sequence model(s) that discover(s) the temporal abstraction structure", eg take any semi-Markov, of Markov-jump-process. However, I agree that this particular version with NN-function approximators / amortized inference is novel and a worthwhile contribution. 2.4) The notation in eqn (1) and (2) looks a bit broken, eg there seems to be $s^i$ missing on the lhs. 2.5) below l81: This process is not exactly the same as the one from eqn (1) and (2) as here the length of the sub-sequence depends on the state as in $p(m_t\vert s_t)$ and not just on the $z_t$.

Reviewer 3



Reviewer knowledge: I am very familiar with variational inference and variational autoencoders. I followed the derivation (in the main paper and appendix) and I believe they are correct. However, I am not very familiar with specific application to temporal data and the further usage in reinforcement learning. Someone familiar with that area should perhaps further comment on connections to prior work (I skimmed the RSSM and the other cited papers for this review). Review summary: interesting and novel work, proposing a new temporal variational model, learning to subsequence data. The experimental work shows the learned subsequences, the models’ ability to predict future frames, and RL performance. My concerns are regarding the forcing of the UPDATE function during testing, as well as the rather limited evaluation and comparisons with other methods (details below). There is also more analysis that can be done, including showing the beneficial effect of modelling uncertainty. Significance: A temporal variational model, which learns sequence structure though latent variables and learned sequence segmentation. The proposed model is principled (can be derived as a bound from the data log likelihood). The idea of learning representations for sequences is very powerful, and can be used for video generation, reinforcement learning, etc. In addition, the paper proposes learning to segment the sequential data into independent chunks, which can be useful for action recognition in videos, text segmentation, or reinforcement learning. The proposed method uses the Gumbel Softmax trick to learn the binary segmentation variables. Note that since this is a biased gradient estimator, the model cannot converge to the true posterior. Originality: I am not an expert on hierarchical sequence modelling, but from what I can tell the proposed method does introduce a novel way to learn hierarchical structure in a probabilistic fashion. Other works either learn the structure in a deterministic fashion (Chung et all, 2016), or they hardcode the structure, or avoid learning the hierarchy all together (Krishnan at all, 2017, Buesing et all 2018a, Chung et all, 2015). The authors focus their empirical comparison with RMSS. Unlike RMSS, the proposed work introduces learning hierarchical structures (through the binary latent variables m which are subsequence delimiters and the latent variables z which encode relevant information regarding a subsequence). The learned variables m and z are missing in the RMSS formulation. However, in RMSS, the authors specifically train the model for long term prediction, via latent overshooting. This is not the case in this work, where the presented work prohibits the use of the COPY operator (lines 113-114), but rather forcing the model to produce a new subsequence at each time step. I am concerned that this is a less principled way to do jumpy imaginatiation, since the model is forced to operate in a situation which it has not seen during training. What if the model would have definitely not started a new sequence at the given time point? In that case, it is forced to generalize outside of its training distribution. There is however an advantage of the way the jumpy navigation is implemented here, and that is efficiency. The proposed method can do jumpy imagination faster than prior work, by forcing the UDPATE operation. Another similar model is VHRED, but this model also does not learn the hierarchical structure (but rather uses a given one). Figure 1 is greatly beneficial for understanding the proposed work. It would also be beneficial to have similar figures for other methods (perhaps in the appendix), as done in Figure 2 in RSSM [Hafner et all]. Experimental results: The experimental work shown exhibits how the model can learn to segment sequences (Figure 3). Since the model learns uncertainty over the learned segments, q(m_t|X), it would have been nice to also see the uncertainty at which the model operates. To show the effect of sequence modelling on navigation, the next experiment shows the ability to model a maze and compares against RSSM. They show the effect of jumpy navigation in Figure 5. The last set of experiments show how the learned model can be used for search in model based RL. The results are obtained only on a simple environment (goal search navigation). Figure 8 shows that the model performs better than RSSM baseline. Here, it would have been nice to see other baselines as well as more complex environments. Note that the model is not trained jointly with the agent (based on the algorithms provided in the appendix), but rather from experience gathered from a random agent. I am concerned that this approach will not scale to hard exploration problems, where a random agent will not be aware of large parts of the environment. Experiments I would have wanted to see: * Something outside the image domain (this has been done in other related works. Example: sequence modelling). See Chung et all, 2016 for examples. * Experiments which exhibit the importance of stochasticity (through a direct comparison with HMRNN). * RL experiments which are directly comparable with prior work. The Goal oriented navigation task is not present in the RSSM paper, so there is no direct baseline to compare to in order to assess the baseline was tuned correctly. * Further RL experiments, for standard, well known, tasks. Nice extra experiments to have: * Showing the generalization effect of the model when forcing the update operation during jumpy evaluation on more than one domain (beyond Figure 5). * [Qualitative and quantitative analysis] What is the effect of the gumbel softmax annealing on the learned subsequence delimiters? Readability: The paper is readable and clear, but I believe certain parts of the appendix need to be moved in the main paper. I am not referring to the derivations themselves, but the overall structure of how the KL term looks like. In that derivation in the appendix, certain notations are being introduced such as (q(s_t)). I suggest to the authors to keep the original notation, such as q(s_t| z_t, s_{t-1}, m_t). While more verbose, it is clearer. The equations can be split on multiple lines. Figure 6 would benefit from further description. Reproducibility: The paper does not provide code but provides the required details in the appendix. I believe the paper is clear enough that the model could be reproduced. However, there are not many details about the provided baseline. Details of the hyperparameter tuning procedure are lacking.

[Author Response · NeurIPS 2019]

[REVIEWER #1] **Long-term dependencies and generative likelihood:** We computed the estimated negative log probability on a test set of Bouncing ball dataset using importance sampling with the learned encoders (500 samplings). The results are RSSM: 3.518 and HRSSM: 3.601. Although it is true that we could expect the proposed model to capture better long-term dependency, we would like to emphasize that our main focus in this work is obtaining *structure* which is interpretable, stochastic, and temporally hierarchical. As such, our main metric is also the quality of the structure, not the accuracy of the generation like language modeling. We are actually satisfying with the fact that we obtain such structure without sacrificing the likelihood performance (of course, it would have been even better with better likelihood performance.) This seems somewhat similar to the fact that discrete latent variable generative models usually do not perform better than its continuous counterpart, but more interpretable. It is in fact not clear whether this learned structure should improve the likelihood performance as well because, unlike other architecture learning problems like NAS, our problem imposes particular temporal hierarchy structures, and thus severely constraints the model space. **Investigation over latent spaces:** We agree. We will add more analysis such as Fig 1 where the temporal abstraction latent $z$ has the subsequence-level context such as *color, direction* and *length*. Fig 2 shows how the subsequence are generated from the same $z$ (but different $s, m$) and the *velocity* of ball and the *length* of subsequence can be varied.

Figure 1: Subseq. with different $z$

[REVIEWER #2] **Interpretable results without color change:** Yes. Because a random color is selected at bouncing from a color set of size 3 including the current color, actually it does not change its color with probability $1/3$. Even in this case, we observe the model cut the segment at bouncing the walls. **Quantitative comparison to a baseline:** This is described above (L1-13) with some explanations. **Maze without action-conditioning:** Yes, we initially also trained the model without action-conditioning and it showed a very similar segmentation result. Due to space limitation, we couldn't include this result, but we will add this result in the Appendix of the camera-ready. We agree that actions and observations can both affect the structure of the segmentation, particularly in a complex way if they are not consistent each other. **RSSM baseline:** RSSM is a single layer version of our model. It is implemented by using the same architecture as the observation-level of our HRSSM and by removing conditioning on $z$ and $m$ (L118-119 in the paper). **Training variation:** The training curve changes rather highly during the early stages of the training as the model searches for a stable temporal structure from $q(M|X)$. It would have been more stable if the temporal structure was given or fixed like VHRED instead of learning it as we do. **Prior over segments:** Our prior is designed to regularize $q(M|X)$ to avoid *over-* and *under-segmentation*. This is done not by explicitly changing $M$ or $q(M|X)$ but by regularizing it through KL and generation terms. That is, if $q(M|X)$ assigns segments that exceeds the limit defined by the prior, it will lead to lower ELBO via the KL term. We, however, agree that more explicitly controlling the posterior class with the segment limits is worth to try. **Independence of $M$ binary indicators:** We agree that giving more structure and conditioning to $q(m_t|X)$ is also worth to try. Nonetheless, we would like to say that $m_t$ is not fully independent but is independent conditionally after observing $X$. Thus, we believe that, although it is somewhat indirect, each $m_t$ can still see the global temporal dependencies by observing the full sequence $X$.

[REVIEWER #3] **Forcing the model to produce a new subsequence at each time step:** Although this is used during training, at test time, we only use the UPDATE operation at the $z$ level without generating subsequences. In experiments, we show that this sequence of $z$'s are good abstract representation of the future and we believe that this is a principled way of performing jumpy imagination in the sense that the formulation is based on the (recurrent) abstract state-space transition model. Also, like other works on imagination-based planning or RL agents, we assume that the test time environment is similar to the training environment so that the learned environment model is useful at test time. So, we do not expect the model to generalize when the test time environment is much different from training distribution. That is, in a new environment, we should not rely on empirically-learned (inductive) imagination until it collects and completes learning from the new environment. **Importance of stochasticity** In our experiment on the navigation task, we observed that we can actually rollout multiple future imaginations from the same state by sampling multiple rollouts. This would not be possible in HMRNN as its rollout is deterministic. This result is obtained when we train the model without action-conditioning. With action-conditioning, we found the model uncertainty reduces and not generate various futures, which is what can be expected. We will add this result in the camera-ready. **Outside image domain:** We agree that applying the proposed model to other domains like text or speech signal would be interesting. For this work, as our main focus was to apply it for agent learning, we put our priority on the image domain and the planning task. **RL experiments which are directly comparable with prior work.** Due to limited time and as our focus was on the structure learning which can also be evaluated by the planning agent without RL, in this work we could not highly prioritize the RL experiment. We, however, agree that it will be helpful in making the paper more complete. **Gumbel softmax:** We used temperature annealing (from $\tau = 1.0$ to $\tau = 0.5$). Without annealing, it was unable to train the model as the entropy of $q(M|X)$ didn't properly decrease and showed meaningless random segmentation.

Figure 2: Subseq with same $z$

[Meta-Review · NeurIPS 2019]

The paper proposes a hierarchical recurrent state space model that can infer latent temporal structure and perform stochastic state transition hierarchically. The idea of jumpy navigation is quite novel and has potential to be adopted and inspire further research. However, the experimental evaluation is a bit weak, even after the additional experiments in the rebuttal.